# Inferring Versatile Behavior from Demonstrations by Matching Geometric Descriptors

**Niklas Freymuth**[1*]    **Nicolas Schreiber**[1]    **Philipp Becker**[1]    **Aleksandar Taranovic**[1,2]

**Gerhard Neumann**[1]

[1]**Autonomous Learning Robots**
Karlsruhe Institute of Technology
Karlsruhe, Germany

[2]**Bosch Center for Artificial Intelligence (BCAI)**
Renningen, Germany

**Abstract:** Humans intuitively solve tasks in versatile ways, varying their behavior in terms of trajectory-based planning and for individual steps. Thus, they can easily generalize and adapt to new and changing environments. Current Imitation Learning algorithms often only consider unimodal expert demonstrations and act in a state-action-based setting, making it difficult for them to imitate human behavior in case of versatile demonstrations. Instead, we combine a mixture of movement primitives with a distribution matching objective to learn versatile behaviors that match the expert's behavior and versatility. To facilitate generalization to novel task configurations, we do not directly match the agent's and expert's trajectory distributions but rather work with concise geometric descriptors which generalize well to unseen task configurations. We empirically validate our method on various robot tasks using versatile human demonstrations and compare to imitation learning algorithms in a state-action setting as well as a trajectory-based setting. We find that the geometric descriptors greatly help in generalizing to new task configurations and that combining them with our distribution-matching objective is crucial for representing and reproducing versatile behavior.

**Keywords:** Imitation Learning, Versatile Skill Learning, Distribution Matching

## 1 Introduction

Imitation Learning (IL) [1, 2, 3] from human demonstrations is challenging as humans often solve tasks in versatile ways. Even the same person might solve a task differently when confronted with it multiple times. Behavior can be versatile in terms of task-level decisions, such as planning a route to a target, and individual actions, such as randomly pausing during a movement to think about the next steps. Most recent IL approaches [4, 5, 6, 7] model the behavior in state-action space using Gaussian policies, which assumes the behavior is unimodal and cannot capture this planning versatility. Yet, this assumption is violated by most human expert datasets, often causing poor generalization to human demonstrations [8]. Additionally, IL approaches often need immense amounts of data to achieve generalization [9, 8]. We tackle these challenges by a feature matching approach to IL that is able to generalize to novel contexts from a small amount of expert demonstrations. Such contexts are (typically low-dimensional) vectors that describe a specific task configuration within a family of related tasks, such as e.g., the coordinates of goal locations to reach. The resulting approach, Versatile Imitation from Geometrically Observed Representations (VIGOR), models distributions that match expert trajectories in terms of concise geometric behavioral descriptors. VIGOR represents distributions using Gaussian Mixture Models (GMMs) over Probabilistic Motion Primitives (ProMPs) [10], and utilizes descriptors that are designed to abstract away from concrete contexts and thus facilitate generalization to novel contexts in a sample-efficient way. An example of such descriptors would be geometric features in the form of (only) the distance between a target

---

[*]correspondence to `niklas.freymuth@kit.edu`

6th Conference on Robot Learning (CoRL 2022), Auckland, New Zealand.

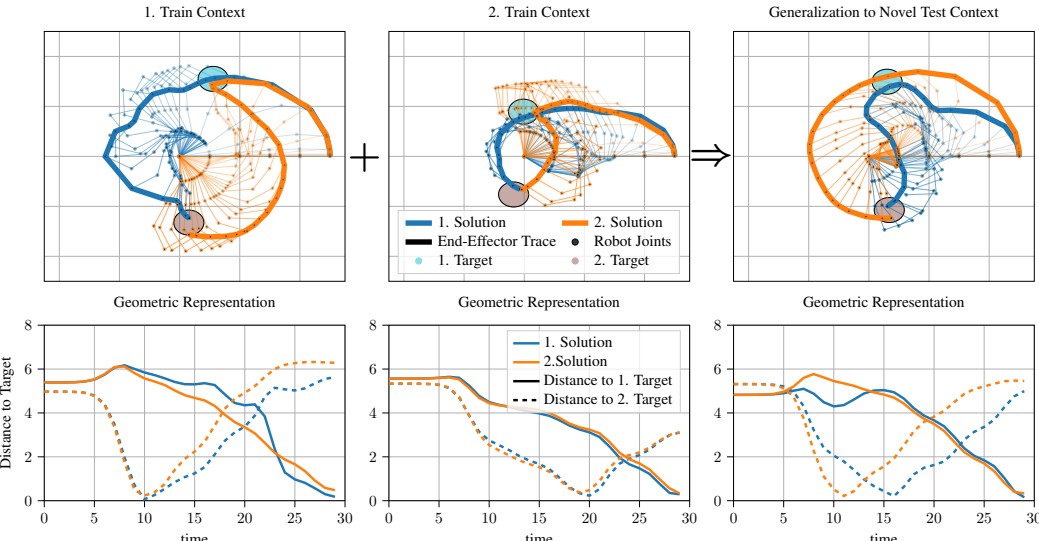

Figure 1: End-effector traces (Top) and distances to target centers (Bottom) for a planar point-reaching task on 3 different contexts. (Left, Middle) Human experts solve the task by reaching the blue target and ending their trajectory in the grey one. (Right) VIGOR matches distributions of behavioral descriptors, such as the depicted target distances, of these demonstrations. This produces versatile behavior in unseen contexts, such as new target positions. Depicted are 2 different component means from a trained GMM Policy. For each context, we show the joint configurations for the first trajectory in blue and the second trajectory in orange.

and the robot's end-effector, plus features that denote the smoothness of the robot. While similar behavioral descriptors have found use in Inverse Optimal Control [11, 12] to recover cost functions from demonstrations, common IL approaches instead use state representations that include both relative and absolute information [5, 13]. The exact form of this set of behavioral descriptors is task-dependent and allows the user to include domain knowledge. Figure 1 shows how VIGOR works on an exemplary reaching task. For training the GMMs, we rely on recent approaches [14, 15, 16] for distribution matching and Variational Inference. For inference, we train individual GMMs for each training context and an arbitrary number of test contexts.

We evaluate VIGOR on a suite of versatile robotic tasks, where demonstrations are collected from human experts through teleoperation. We compare our method to standard Behavioral Cloning (BC) [17], BC with a GMM policy [13, 18], Generative Adversarial Imitation Learning (GAIL) [5] and Inverse Reinforcement Learning (IRL) techniques [7]. We find that these methods either fail to learn useful behavior because they average over multiple solutions, or that they are unable to capture the full versatility of the demonstrations. Contrary to this, VIGOR accurately models highly versatile behavior from already a small number of trajectories demonstrated by human experts. We perform extensive ablation studies to showcase the importance of different parameters and design choices. Datasets and code can be found at https://www.github.com/NiklasFreymuth/VIGOR.

To summarize, our list of contributions is as follows:

- We infer *multi-modal distributions* of desired trajectories from highly versatile human expert trajectories by matching distributions over behavioral descriptors between learner and expert.

- Our approach is, to the best of our knowledge, the first Adversarial Imitation Learning method to utilize *concise behavioral descriptors* to facilitate generalization to novel contexts from already a small number of demonstrations.

- We conduct *experiments with human demonstrations* in simulation and on a real robot and find that VIGOR accurately models highly versatile human behavior, outperforming various Imitation Learning baselines.

## 2 Related Work

**Imitation Learning for Skills.** A common way to represent skills over trajectories is via Movement Primitives (MPs) [19, 10]. While learning individual MPs from human demonstrations is a simple regression problem, modeling versatile behavior requires a more sophisticated model that can represent such multi-modality, e.g., a mixture model. Using Expectation Maximization (EM) [20], both [21] and [22] fit a single GMM over MP parameters of multiple demonstrations for different contexts and use them to generalize to novel contexts by conditioning. Yet, as they only fit a single GMM for all contexts, they do not explicitly focus on representing versatility. Recent work [23] instead learns a mixture over contextualized MPs using Gaussian Mixture Regression. This has been extended to non-linear relations between MP parameters and context [18] by using Mixture Density Networks (MDNs) [24]. While the above approaches work for tasks with a small number of modes, optimizing MDNs can be challenging in the case of many modes in the demonstrations. Osa et al. [25] use planning algorithms to reproduce human demonstrations in a trajectory-based setting. They tune the parameters of a cost function such that the planned trajectories match the demonstrations and use the extracted cost function to guide the trajectory optimization process. Yet, this approach is limited to uni-modal demonstrations and requires engineered cost functions.

**Imitation Learning by Distribution Matching.** Classical IL approaches [26, 27] employed feature expectation matching to learn a policy from behavioral descriptors of expert demonstrations. Similarly, a recent body of work [11, 12] in Inverse Optimal Control utilizes geometric behavioral descriptors that are similar to ours to learn cost functions for manipulation tasks from a few demonstrations. Whereas these approaches linearly match the moments/expectations of their features, our method instead matches a versatile distribution over non-linear features using the reverse Kullback-Leibler Divergence (KL) [28]. Directly matching the distribution rather than its moments means that our method can represent complex and multi-modal distributions. More recently, a new class of distribution matching approaches has gained popularity with the advent of Generative Adversarial Nets [29]. Starting from GAIL [5], a whole class of distribution matching based IL approaches was developed [4, 30, 31, 32, 33, 34]. All of these approaches commonly work in a step-based setting and minimize some divergence between the state-action occupancy or state marginals. They generally do not consider versatile behavior, and additionally need to interact with the environment during training in order to generate states to match. We instead work in a trajectory-based setting , which allows us to abstract away the environment's dynamics and learn entirely offline, i.e., without any environment interactions. Common methods in this setting usually employ a maximum likelihood approach, which, as previously discussed, performs poorly on versatile data. As a solution to this problem, Becker et al. [16] propose using the Information-Projection [35] instead, which is able to focus on individual modes of data rather than averaging over all modes of data. This approach has been extended to IRL [36]. Here, we build on these methods, generalizing them to sequential data and geometric behavioral descriptors.

**Comparison-Based Approaches.** Brown et al. [6] perform IRL by utilizing provided rankings of demonstrations to train a discriminator on a comparison-based loss. As the discriminator learns to favor samples with a high rank, it can be used as a reward function after training. This has been extended by Disturbance-based Reward Extrapolation (D-REX) [7], which automatically generates rankings by fitting a BC policy on the demonstrations and subsequently draws samples with different levels of noise from this policy. We adapt D-REX to our trajectory-based setting as a baseline. Similar to VIGOR, this adapted method trains a model on expert demonstrations on training contexts to optimize GMM policies on novel test contexts. However, while D-REX learns a reward function to do so, VIGOR iteratively (re-)trains a discriminator on the expert demonstrations and policy samples.

## 3 Foundations

In this section, we briefly cover the foundations and previous work that our method builds on.

**Probabilistic Movement Primitives.** For a trajectory $\tau = (\tau_1, \cdots, \tau_T)$, ProMPs represent the elements as $\tau_t = \boldsymbol{\Phi}(t)^T \boldsymbol{w}$, where $\tau_t$ represents a vector of the desired joint angles at time-step $t$ [18] Note that $\boldsymbol{w}$ does not depends on $t$ and that $\tau$ can thus be computed at an arbitrary resolution $T$. Here $\boldsymbol{\Phi}(t)$ are time-dependent features, usually radial basis functions centered around different time points, and $\boldsymbol{w}$ denotes the parameter vector. For a single demonstration, the parameters $\boldsymbol{w}$ are fitted using simple linear regression and compactly represent the entire trajectory.

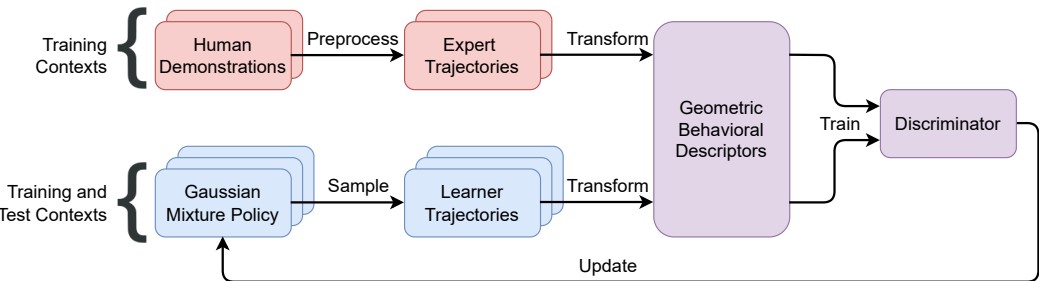

Figure 2: Schematic of VIGOR. We generate a set of expert trajectories from human demonstrations and then transform them into geometric behavioral descriptors. These descriptors are then fed into a discriminator. This process is repeated for samples of a separate GMM policy for each training and test context. We then train the discriminator to distinguish between human and policy trajectories. The geometric descriptors are designed to abstract away from concrete task configurations, causing the discriminator to distinguish how well a trajectory performs rather than which context it acts on. As a result, the discriminator can be used to improve the policies.

**Variational Inference for Gaussian Mixture Models.** We build on a recent class of efficient Variational Inference approaches that allow modeling versatile behavior with GMMs [16, 14]. These approaches use the Information-Projection [35] to minimize the reverse Kullback-Leibler Divergence $\mathrm{KL}\big(q(\boldsymbol{w})||p(\boldsymbol{w})\big)$ between a model $q(\boldsymbol{w})$ and a target distribution $p(\boldsymbol{w})$. In our case, $p(\boldsymbol{w})$ is the distribution of expert trajectories. Arenz et al. [14] introduced an upper-bound objective based on a variational decomposition. The resulting approach, Variational Inference by Policy Search (VIPS) [14, 15], repeatedly solves the objective

$$q\left(\boldsymbol{w}\right) = \operatorname*{arg\,min}_{q(\boldsymbol{w})} \mathbb{E}_{q(\boldsymbol{w},\boldsymbol{z})}\left[\log\frac{\hat{q}\left(\boldsymbol{w}\right)}{p\left(\boldsymbol{w}\right)}\right] + \mathrm{KL}\big(q(\boldsymbol{z})||\hat{q}(\boldsymbol{z})\big) + \mathbb{E}_{q(\boldsymbol{z})}\Big[\mathrm{KL}\big(q\left(\boldsymbol{w}|\boldsymbol{z}\right)||\hat{q}\left(\boldsymbol{w}|\boldsymbol{z}\right)\big)\Big], \quad (1)$$

where $\hat{q}$ denotes the model from the previous iteration. This objective decomposes into individual optimizations for the components and categorical distributions, which Arenz et al. [14] solve using trust-region methods from policy search [37, 38].

**Distribution Matching for Gaussian Mixture Models.** We cannot directly work with VIPS, as we do not know $p(\boldsymbol{w})$ but only have the expert demonstrations. To address this issue, Becker et al. [16] propose using density ratio estimation [39] approaches to approximate the $\log(\hat{q}(\boldsymbol{w})/p(\boldsymbol{w}))$ term in Equation 1, removing the dependency on $p(\boldsymbol{w})$. Specifically, they use logistic regression to estimate the density ratio. The resulting approach, Expected Information Maximization (EIM), resembles Generative Adversarial Nets [29] where the discriminator effectively also estimates a density ratio.

## 4 Inferring Versatile Behaviors by Matching Geometric Descriptors

Our approach estimates versatile behavior in form of GMM distributions over ProMP weights for all training and test contexts of the given task. We note that while this setup requires the test contexts in advance, it does not need interaction with the real environment since the behavioral descriptors are directly computed from the proposed trajectory and its context. To capture correlations in the demonstrations, we utilize full-covariance GMM components. For each planned trajectory, we compute concise descriptors such as distances to key-points that isolate the performance of the trajectory from its context to allow for sample-efficient generalization to novel contexts. We then build on EIM [16] and use a discriminator to infer GMMs from demonstrations by matching the learner's and expert's distributions in the resulting geometric feature space. Towards this end, we generalize EIM to sequential data and modify the policy parameterization to accommodate for our setting. Figure 2 illustrates an overview of our approach. We provide pseudocode in Appendix A.

**Context-Specific Mixture Policies** We want to model the expert's behavior for a given small set of training configurations of the task, denoted as training contexts $\boldsymbol{c}_{\mathrm{train}}$, and a set of unseen test contexts $\boldsymbol{c}_{\mathrm{test}}$. We consider the set of all contexts as $\boldsymbol{c} = \boldsymbol{c}_{\mathrm{train}} \cup \boldsymbol{c}_{\mathrm{test}}$. As context-dependent GMMs[2] are hard

---

[2]Such GMMs could in principle be represented by MDNs. However, attempts with MDN and full-covariance Gaussian components were highly unsuccessful in our experiments.

to learn and we assume a relatively small amount of training contexts, 6 in all our experiments, with multiple demonstrations per training context, we resort to maintaining a single GMM-Policy $q_c(\boldsymbol{w})$ for each context $c$. This non-amortized formulation simplifies the policy updates as we can optimize the policy for each context individually, given the density ratio estimator, using highly efficient, tailored methods for full-rank GMM approximations [14, 15].

**Distribution Matching of Behavioral Descriptors** We assume access to behavioral descriptors $\boldsymbol{O} = f_c(\boldsymbol{w})$ that represent features encoded in the parameter vector $\boldsymbol{w}$ with respect to the context $c$ of $\boldsymbol{w}$. For simplicity, we will assume that the mapping $f_c$ between $\boldsymbol{w}$ and $\boldsymbol{O}$ is deterministic, such that we can easily evaluate $\boldsymbol{O}$ for any desired plan $\boldsymbol{w}$. We want to match the distribution of behavioral descriptors of the demonstrator while optimizing for $q(\boldsymbol{w})$, i.e,

$$q^*(\boldsymbol{w}) = \underset{q(\boldsymbol{w})}{\arg\min}\, \mathrm{KL}\big(q\left(\boldsymbol{O}\right)||p(\boldsymbol{O})\big) = \underset{q(\boldsymbol{w})}{\arg\min}\, \mathbb{E}_{q(\boldsymbol{w})}\left[\int_{\boldsymbol{O}} p(\boldsymbol{O}|\boldsymbol{w})\log\frac{q(\boldsymbol{O})}{p(\boldsymbol{O})}d\boldsymbol{O}\right],$$

where $p(\boldsymbol{O}|\boldsymbol{w})$ is a Dirac delta distribution defined by the mapping $f_c$ and $q(\boldsymbol{O}) = \int p(\boldsymbol{O}|\boldsymbol{w})q(\boldsymbol{w})d\boldsymbol{w}$. As the mapping between $\boldsymbol{w}$ and $\boldsymbol{O}$ is deterministic, the EIM algorithm can be directly applied to this setup with the difference that the discriminator is trained using $\boldsymbol{O}$ instead of $\boldsymbol{w}$. Moreover, while we a learn different distributions $q_c(\boldsymbol{w})$ for each context $c$, we use the same discriminator for all contexts. This way, we can infer distributions $\{q_c(\boldsymbol{w})|c \in \boldsymbol{c}_{\text{test}}\}$, i.e. infer versatile trajectories for unseen scenarios.

**Density Ratio Estimation for Sequential Behavioral Descriptors** As in our case, $\boldsymbol{w}$ encodes the desired trajectory $\tau$, the behavioral descriptors are typically computed per time-step, i.e., $\boldsymbol{O} = (\boldsymbol{o}_1, \ldots, \boldsymbol{o}_T)^T$, where each $\boldsymbol{o}_t$ is a feature vector. To enable the classifier to deal with sequential data, we consider a sequence-to-sequence neural network $\boldsymbol{\phi}(\boldsymbol{O}) = \boldsymbol{\phi}((\boldsymbol{o}_1, \ldots, \boldsymbol{o}_T)^T) = (y_1, \ldots, y_T)^T$. The network receives a sequence of inputs $\boldsymbol{O} = (\boldsymbol{o}_1, \ldots, \boldsymbol{o}_T)^T$ and outputs a sequence of values $(y_1, \ldots, y_T)^T$, where each $y_i$ may depend on multiple $\boldsymbol{o}_j$. In each iteration of our optimization, we (re-)train this network to discriminate between the provided demonstrations $\boldsymbol{O}^{(p)}$ of $\boldsymbol{c}_{\text{train}}$, and an equal number of policy samples $\boldsymbol{O}^{(q)}$ drawn uniformly from $\boldsymbol{c}$. We first consider the case where we want to discriminate full trajectories as in EIM. Denoting the sigmoid function as $\sigma$, the discriminator can straightforwardly be trained on a binary cross-entropy loss of the sum of sequence values $\hat{y} = \sum_{t=0}^T y_t$, i.e., by minimizing

$$\mathrm{BCE}\big(\boldsymbol{\phi}(\boldsymbol{O}), \boldsymbol{O}^{(p)}, \boldsymbol{O}^{(q)}\big) = -\mathbb{E}_{\boldsymbol{O}^{(q)}}\left[\log\left(\sigma\left(\hat{y}\right)\right)\right] - \mathbb{E}_{\boldsymbol{O}^{(p)}}\left[\log\left(1 - \sigma\left(\hat{y}\right)\right)\right], \quad (2)$$

w.r.t. $\boldsymbol{\phi}(\boldsymbol{O})$. Equation 2 recovers the log density ratio $\boldsymbol{\phi}(\boldsymbol{O}) = \log \boldsymbol{O}^{(p)} - \log \boldsymbol{O}^{(q)}$ at convergence [39]. However, this cost function only provides a single training sample per trajectory and is therefore hard to use for a small number of demonstrations. By utilizing the sequential structure of the trajectories, the classification error can be computed for each time-step, i.e. we minimize

$$\mathrm{BCE}_{\text{step}}\big(\boldsymbol{\phi}(\boldsymbol{O}), \boldsymbol{O}^{(p)}, \boldsymbol{O}^{(q)}\big) = -\mathbb{E}_{\boldsymbol{O}^{(q)}}\left[\sum_{t=0}^T \log\left(\sigma\left(y_t\right)\right)\right] - \mathbb{E}_{\boldsymbol{O}^{(p)}}\left[\sum_{t=0}^T \log\left(1 - \sigma\left(y_t\right)\right)\right] \quad (3)$$

w.r.t. $\boldsymbol{\phi}(\boldsymbol{O})$. Finally, we train an ensemble of discriminators instead of a single one, using their average logit as the log density ratio estimate for the policy updates. For the network architecture, we find that simple $2-4$ layer $1d$ convolutional neural networks ($1d$-CNNs) work best in our experiments. We compare this choice to other sequential network architectures the ablations in Appendix D.

**Geometric Behavioral Descriptors** VIGOR utilizes geometric descriptors that abstract away from a concrete task configuration to generalize to new configurations. To this end, we encode the current state along the desired trajectory with respect to relative geometric features rather than absolute values. The resulting geometric descriptors cause trajectories of similar performance to appear similar in feature space regardless of their context, allowing for sample-efficient generalization to novel contexts. We note that such a descriptor space can often be straightforwardly constructed from the geometry of the task by composing distances of the end-effector to key-points of the objects in a scene. For example, in object manipulation, this space can be composed of distances of the end-effector to the corners of a box to push.

## 5   Experiments

All experiments use human demonstrations. The tasks were selected due to their highly multi-modal nature and because they allow for an easy collection of human demonstrations. We instructed the

demonstrators to solve the task in varying ways to create versatile solutions. As a preprocessing step, we fit a ProMP for each human demonstration and filter the resulting ProMPs such that only successful demonstrations remain. This step ensures that the human demonstrations can be imitated by the learner and conveniently allows to compress all demonstrations to a fixed length in a principled fashion. We use the resulting preprocessed expert demonstrations for all experiments unless otherwise stated. Similarly, unless noted otherwise, all experiments use 6 training and 6 test contexts, and 5 GMM components for each configuration and method. Experimentally, this number of components is sufficient for matching the distribution of expert descriptors for the considered tasks, whereas more components generally only lead to spurious improvements. We experiment with less components in the ablations in Appendix D. Unless noted otherwise, we do *not* train the categorical distribution of the GMM components, using a uniform distribution instead. We evaluate the GMM policies using a number of samples for each component of each *test* context, and then rank these components according to the average performance of their samples. We then report statistics of the best component of the resulting value over the random seeds. Note that reporting the best component leads to a fair comparison to unimodal approaches. For baselines that do not use GMM policies, we instead draw samples of the trained policies for each test context and average the performance of all of them. For each experiment, we report a measure of how well the task is performed in the main paper, and additional success rates in Appendix C. We repeat each experiment for 10 random seeds. Appendix E shows all hyperparameters and training details.

**Baselines.** We compare VIGOR to a number of IL baselines, using three state-action baselines, namely GAIL [5], behavioral cloning (BC(S)) and behavioral cloning with a GMM policy (BC-GMM(S)) [13]. For all experiments, the state-action baselines receive geometric descriptors of the current state and a time-step as state information, and output the velocities of the robot joints as actions. Further, we also consider three trajectory-based methods. These are standard behavioral cloning (BC(T)), behavioral cloning with a GMM policy (BC-GMM(T)) (see e.g., Zhou et al. [18]), and a modified version of D-REX (EM+D-REX). The latter is chosen due to its similarities to VIGOR, which are detailed in Appendix B.1. We refer to Appendix B for details on the baselines and to Appendix C for further results on all experiments described below.

**Planar Reacher.** First, we consider the introductory task shown in Figure 1. In this task, the goal is to reach an intermediate target area with the end-effector of a 5 Degrees of Freedom (DoF) robot with joints of length 1 before ending the trajectory in a final target area. The geometric descriptors for this task are given by the euclidean distance of the end-effector to both targets, paired with the average velocity and acceleration of the joints to encode the smoothness of the motion. This yields a total of 4 features per time-step. The left and middle panels of Figure 1 depict example demonstrations for two training contexts (top) and the corresponding distances between the end-effector and targets over time (bottom). We explore other choices for the geometric descriptors in the ablations in Appendix D. For evaluation, we use the *Distance to Boundaries* of the target areas. For the first target, we consider the minimum distance of the end-effector to the boundary; for the second target, we consider the distance at the last time-step. We train all methods on 5 demonstrations per training context. The left of Figure 3 shows results on test contexts. We find that only VIGOR and EM+D-REX can consistently get close to both targets. Yet, EM+D-REX can only learn a single mode and fails to reproduce the versatility of the demonstrations. We investigate this in more detail in Appendix C.1.

**7-D Panda Reacher.** We repeat the above point-reaching task using a 7 DoF Franka Emika Panda robot and 3-D intermediate and goal positions. Since we do not care about the end-effector's rotation, the last DoF can be ignored, leading to an effective action dimension of 6. We use 8 basis functions per action dimension for the MPs, leading to a total of MP dimension of 48 and thus a high-dimensional problem space. The expert demonstrations are collected using a teleoperation setup with a virtual twin that records joint values over time. Appendix C.2 explains the setup in more detail. We use the same geometric descriptors as for the planar reacher task. The right of Figure 3 shows target distances on test contexts. Similar to the previous task, only VIGOR and EM+D-REX can consistently get close to both targets while the other methods regularly fail to pursue at least one of the two targets.

**Box Pusher.** We also evaluate our approach on a contact-rich robot manipulation task. In this task, a simulated Franka Emika Panda robot has a rod attached to its end-effector and needs to use it to push a rectangular box to a given goal position and orientation. The box always starts at the same position, and the task context is given by the desired $(x, y)$ translation of the box, plus its desired rotation in degrees. For simplicity, the task is learned in task space with a fixed height. The demonstrations contain the starting position of the robot as well as its trajectory. Here, the solutions are versatile

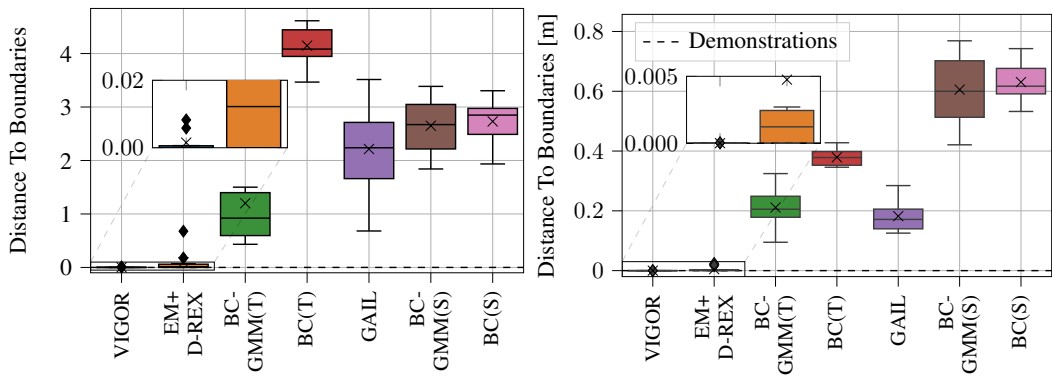

Figure 3: Mean target distance of samples from the best policy component on the Planar (left) and Panda (right) Reaching tasks for test contexts. We find that a trajectory-based setting and the use of mixture policies help performance, as can be seen from VIGOR, EM+D-REX, and BC-GMM (T). Similarly, using a discriminator to iteratively optimize the policies is advantageous on the reaching tasks, as seen from VIGOR and GAIL. Finally, the clear distribution matching objective of VIGOR and EM+D-REX seems crucial for imitation from a low number of versatile demonstrations. VIGOR uniquely combines the above properties and is able to reliably reach both targets for both tasks.

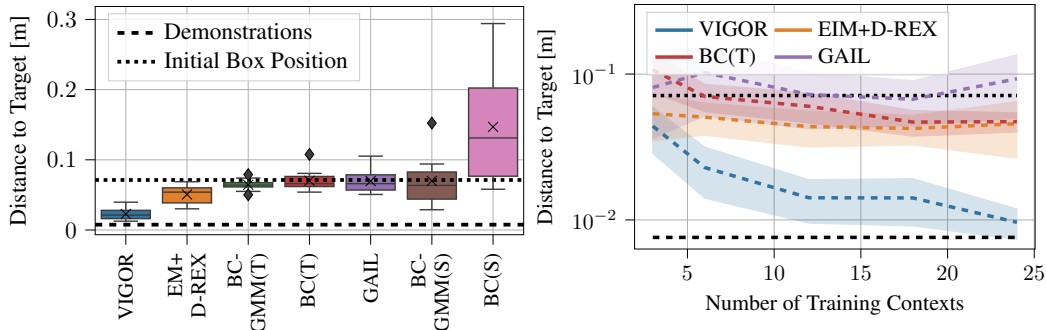

Figure 4: Mean target distance of samples from the best policy component on the Box Pushing task for test contexts. The dotted line denotes the average corner distance between the initial and the desired box position. The dashed line shows the performance of the human demonstrations. (Left) VIGOR is able to consistently push the box to the right configuration. Opposed to this, the other methods only marginally improve over the initial box positions, and in some cases perform worse than it. (Right) We find that VIGOR benefits most from additional training contexts, almost matching the performance of the demonstrator on unseen contexts for 24 training contexts.

because the box can be pushed both from the inside and outside. The geometric descriptors consist of the $(x, y)$ distances of the end-effector to the *initial* and *desired final* position of the corners of the box. To facilitate generalization between contexts, we translate the basis of these distances such that the desired final box aligns with the origin of the coordinate system. We additionally add the end-effector velocity and acceleration as well as the time-step, resulting in 19 features per time-step. This geometric description of the task allows for IL without simulating or explicitly modeling the box, significantly speeding up the training process and facilitating generalization to both new contexts and real-world robots. We use 3 demonstrations per training context and evaluate the average distance of the corners of the final box to that of the *desired final* position. The contact with the box causes a mismatch between a planned trajectory and its execution on the robot, which we illustrate in Appendix C.3. Results are shown in the left of Figure 4. Additionally, the right of Figure 4 shows how the performance varies when changing the number of training contexts. Finally, we illustrate how the planned trajectories of VIGOR perform on a real robot. To this end, we create a real-world replica of the simulated box that the robot manipulates. Rollouts for different policy components of VIGOR trained on 24 training contexts on an exemplary test context are shown in Figure 5.

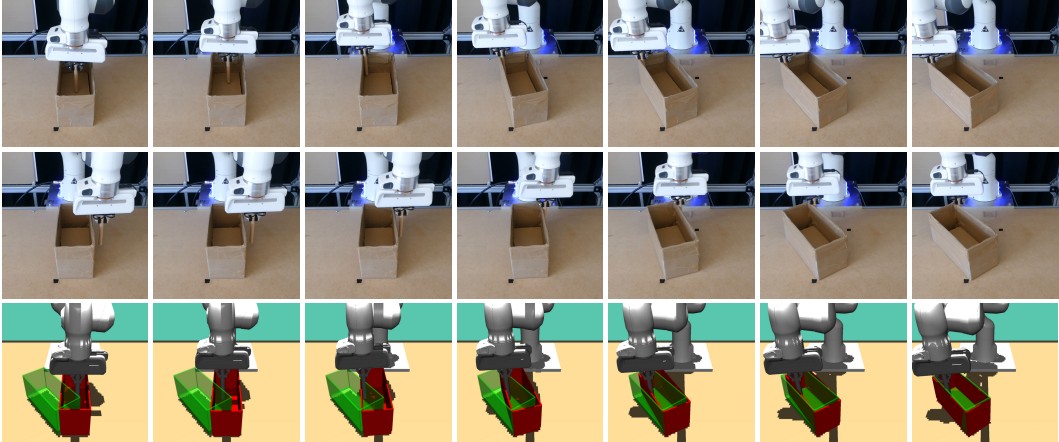

Figure 5: Trajectory executions of three component means of a VIGOR policy trained on $24$ training contexts for the same test context. The first two rows show a real Franka Panda executing the planned end-effector trajectories. The last row shows the same setup in the simulation, where the target box is overlaid in green. All three executions push the box in different ways, capturing the versatility of the human demonstrations; while the robot in the first row pushes the box from the inside, the robot in the second row pushes it from the outside. The third trajectory also pushes from the inside but is much farther away from the corner of the box.

**Ablations.** We conduct extensive ablation studies to investigate how our design choices affect our approach. We find that a concise choice of behavioral descriptors, a preprocessing of the expert demonstrations to ProMPs, and a $1d$-CNN discriminator are crucial for performance. We also notice that the policy benefits from additional components and that we see modest improvements for an ensemble of discriminators and the stepwise loss proposed in Equation 3. We refer to Appendix D for the full results and more thorough ablations of both VIGOR and EM+D-REX.

## 6 Limitations

**Scale.** We currently train and maintain a separate GMM over ProMP parameters for each context, leading to linear space and time complexity w.r.t. the total number of contexts. One way to address this challenge would be to instead train a joint Mixture Density Network $q(\boldsymbol{w}|c)$ over all contexts, which is left for future work. Additionally, representing trajectories with a single ProMP limits them to single smooth movements. To alleviate this, MP *chaining* [40, 41, 42] can be straightforwardly integrated into our approach to allow for the representation of more complex movements

**Geometric Descriptors.** VIGOR facilitates generalization to novel task configurations by using geometric descriptors. While this allows the user to include domain knowledge, it also requires a clear idea about which aspects of the task are important. Instead, geometric descriptors could be automatically extracted from the task for example by using NNs processing point clouds [43, 44].

**Re-training.** Finally, our method assumes that the test configurations $\boldsymbol{c}_{\text{test}}$ are known in advance. Since the test configurations are optimized jointly with the training configurations, the only way to integrate new test configurations is to re-train the algorithm. In the future, we want to alleviate this issue by recovering a reward representation from ranked intermediate policies of the given training configurations, essentially combining our approach with some of the ideas of D-REX.

## 7 Conclusion

We proposed VIGOR, a novel feature-matching approach to Imitation Learning in environments with versatile solutions. Utilizing a combination of movement primitives, mixture policies and matching geometric behavioral descriptors, our method can closely imitate the behavior distribution of human experts from a few demonstrations. We show the effectiveness of our approach on a suite of challenging robot coordination tasks. The results show that VIGOR is able to closely match the distribution of the demonstrator, outperforming the chosen baselines in all considered settings.

**Acknowledgments**

NS and GN were supported by the Carl Zeiss Foundation under the project JuBot (Jung Bleiben mit Robotern). The authors acknowledge support by the state of Baden-Württemberg through bwHPC.

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
