# OpenReview forum: "Inferring Versatile Behavior from Demonstrations by Matching Geometric Descriptors"
_robot-learning.org/CoRL/2022/Conference — CoRL 2022 Poster_

### Official Review · Reviewer_TvrN · 2022-07-31

**Originality:** Good
**Technical Quality:** Good
**Clarity Of Presentation:** Good
**Impact:** 2

**Recommendation:**

Weak Accept: I recommend accepting the paper, but will not argue for my recommendation if the majority of other reviewers have a different opinion.

**Summary:**

The paper is concerned with the problem of using imitation learning (IL) to find behavior policies that allow machines to accomplish tasks when the set of demonstrations that should be imitated are "versatile," meaning that it contains multiple ways to accomplish a particular task. The paper proposes a new IL method called VIGOR which is based on probabilistic motion primitives and distribution matching over "geometric features," and presents an evaluation of this new method for both simulated and real robot arm tasks.

**Issues:**

(see weaknesses identified above)

**Quality Of The Limitations Section:**

Limitations are addressed clearly

**Reviewer Expertise:**

3: The reviewer is fairly confident that the evaluation is correct

**Robotics Focus:**

Sufficient demonstration on hardware

**Strengths And Weaknesses:**

(+) The paper does an excellent job of illustrating the problem of "versatile" demonstrations and performs experiments using real human demonstrations, which is compelling.

(+) Experiments are performed on both simulated and real robots.



(-) It's not clear to me how strong the contribution of the paper is here. The introduction of the paper makes it seems as though the authors are claiming that the use of geometric features constitutes the novelty of the proposed method, but merely representing the state features of a problem in terms of, eg, distance to goal, seems commonplace in the community. If the authors disagree, could they provide citations to important recent papers that do _not_ do this? Additionally, from the ablation studies, it also seems important for VIGOR that the demonstrations be "filtered" by first fitting a ProMP; are the authors claiming that doing this is also a novel part of their approach, or has this been done in other work? An explicit list of claimed contributions—along with justifications for why these are indeed contributions—would go a long way to improving the paper.

(-) The technical discussion needs to be improved for clarity. For example, Section 3 starts off by defining a trajectory $\tau = {\tau_1, …}$, but never actually defines for the reader what type of mathematical object $\tau_t$ is (eg, is it a scalar? a vector? what specifically does it encode? a state?). The same goes for the precise meaning of $\phi$ and $O$. This missing information makes it hard for me to evaluate the proposed algorithm.

**Summary Of Recommendation:**

The problem of performing imitation learning in the presence of "versatile demonstrations is interesting and the paper's illustration of this phenomenon, along with interesting experiments is good. However, the specific contributions to the literature are unclear to me and the technical discussion also needs to be improved for clarity.

POST-DISCUSSION UPDATE: During the discussion phase, the authors addressed several of my questions, and made corresponding revisions to the paper. I now think I have a better understanding of the novelty of the paper here, and am raising my score to "Weak Accept." However, I still think that the authors can do a better job in the presentation here by more explicitly focusing on the truly novel portion of their algorithm as opposed to the features that it relies on having access to. I understand this is an important point, but the authors have provided very little here to the community regarding how to obtain these features.

---

> ### Author Response · Authors · 2022-08-22
> **Reply to Reviewer TvrN (Part 1/2)**
>
> Please note that this reply has been split into two parts due to character limits. (Part 1/2)
>
> We thank the reviewer for their constructive comments and detailed feedback, particularly for suggesting a clear list of contributions as well as clarifications in the technical discussion. In the following, we want to address the individual concerns mentioned by the reviewer. We will upload a revised version of the paper in due time and look forward to further comments from the reviewer.
>
> > (-) It's not clear to me how strong the contribution of the paper is here. The introduction of the paper makes it seems as though the authors are claiming that the use of geometric features constitutes the novelty of the proposed method, but merely representing the state features of a problem in terms of, eg, distance to goal, seems commonplace in the community. If the authors disagree, could they provide citations to important recent papers that do *not*
>  do this?
> >
>
> We fully agree with the reviewer that expressing the novelty of the chosen behavioral descriptors can be improved in the paper. To clarify, the novelty of these features is that they are chosen to abstract away from individual contexts for the express purpose of facilitating generalization to novel contexts from few expert demonstrations. The features are constructed such that well-performing trajectories look similar in feature space regardless of their context, while being dissimilar to poor-performing trajectories.This is done by choosing concise geometric features relative to the current context.
>
> Comparing this to previous work, we find that often a mixture of relative and absolute features is used. For example, [1] combines proprioceptive observations of the robot with task-dependent positions of objects and distances between the robot and these objects. Especially adversarial methods are often trained directly on either states or state-action pairs [2, 3] of their respective environments, which usually include extensive state information. None of these methods explicitly use their state features to improve generalization. In our setup, this additional information would allow the discriminator to distinguish trajectories by context rather than performance, leading to worse generalization.
>
> > Additionally, from the ablation studies, it also seems important for VIGOR that the demonstrations be "filtered" by first fitting a ProMP; are the authors claiming that doing this is also a novel part of their approach, or has this been done in other work?
> >
>
> We find that fitting a ProMP on each expert demonstration is a necessary preprocessing step as it slightly smoothes the expert demonstrations and prevents the discriminator from focusing on parts of the (unprocessed) expert trajectories that can not easily be imitated by a ProMP, e.g., non-smooth parts of the trajectory. We note that other methods such as [4] also fit ProMPs on their expert demonstrations. However, this is done because these methods directly learn in the ProMP feature space $w$, i.e., they usually do not generate trajectories $\tau$ from these features. We do *not* claim this preprocessing step as a novelty.
>
> > An explicit list of claimed contributions—along with justifications for why these are indeed contributions—would go a long way to improving the paper.
> >
>
> Our list of contributions is as follows:
>
> - We infer desired trajectories from already a small number of multi-modal human expert trajectories by matching a distribution over behavioral descriptors of the desired trajectories to the distribution of the expert.
> - While our approach is similar to existing Adversarial Imitation Learning approaches in the sense that it uses a discriminator to update the policy, to the best of our knowledge our approach is the first approach that utilizes behavioral descriptors instead of state-action distributions to generalize to novel contexts.
> - Other adversarial distribution matching methods can only obtain unimodal behavior and are matching the state-action distribution which does not generalize to new contexts. We detail this in the "Imitation Learning by Distribution Matching" paragraph of the related work section of the paper. Our approach is also different from methods that use mixture policies to match the moments of either state-action pairs [1] or full trajectories [4] rather than using behavioral descriptors.
> - We conduct experiments with human demonstrations as well as on a real robot, performing extensive ablation studies to showcase the importance of different parameter choices and design decisions. The datasets and code will be published with the camera-ready version.
>
> We will add a list of these contributions and their justifications to the revised paper.

---

> > ### Author Response · Authors · 2022-08-22
> > **Reply to Reviewer TvrN (Part 2/2)**
> >
> > Please note that this reply has been split into two parts due to character limits. (Part 2/2)
> >
> > > (-) The technical discussion needs to be improved for clarity. For example, Section 3 starts off by defining a trajectory $\tau=\tau_1, …, $ but never actually defines for the reader what type of mathematical object $\tau_t$ is (eg, is it a scalar? a vector? what specifically does it encode? a state?). The same goes for the precise meaning of $\phi$ and $\mathbf{O}$. This missing information makes it hard for me to evaluate the proposed algorithm.
> > >
> >
> > Thank you for pointing out that this technical discussion needs clarification.
> >
> > - The paper follows [5], defining $\tau_t$ as the vector of desired joint angles at timestep $t$. We will add this in the revision for clarification.
> > - The behavioral descriptor $O$ can in principle be any function of movement primitive parameters $\mathbf{w}$, i.e., $\mathbf{O}=f(\mathbf{w}$. Since $w$ represents the desired trajectory, we make use of a decomposition $\mathbf{O}=(\mathbf{o}_1, \dots, \mathbf{o}_T)^T$ of features over time. Here, each $\mathbf{o}_t$ is a vector containing $4-19$ (geometric) descriptors for the timestep $t$ that is usually computed based on the joint angles $\tau_t=\mathbf{\Phi}(t)^T\mathbf{w}$. We set $T=30$ for the Planar Reacher and $T=50$ for the other tasks. These parameters are detailed in Appendix E.1. We note that the function $f(w)$ also depends on the context $c$ of $w$, which we omitted to avoid clutter. For the revision, we will clarify this relationship and add an explicit dependence by changing the notation to $f_c(w)$.
> > - The discriminator $\phi$ is some function that can be trained to classify whether a behavioral descriptor $\mathbf{O}$ belongs to an expert demonstration or the learner policy. Since we assume that $\mathbf{O}=(\mathbf{o}_1, \dots, \mathbf{o}_T)^T$, we choose $\phi$ to be a sequential neural network. In our experiments, $\phi$ is a simple $2-4$ layer $1d$-CNN with $32-128$ neurons/channels per layer and a kernel size of $5-7$. We compare this choice to stepwise MLPs and LSTMs in the appendix in Figure $11$. Detailed network hyperparameters can be found in Appendix E.2.
> >
> > We will clarify these points in the revised version of the paper.
> >
> > [1] Mandlekar et al. “What Matters in Learning from Offline Human Demonstrations for Robot Manipulation”, *5th Annual Conference on Robot Learning*, 2021
> >
> > [2] Orsini, Manu, et al. "What matters for adversarial imitation learning?." *Advances in Neural Information Processing Systems*, 2021
> >
> > [3] Ho, Jonathan, and Stefano Ermon. "Generative adversarial imitation learning." *Advances in neural information processing systems*, 2016
> >
> > [4] Zhou, You, et al. "Movement primitive learning and generalization: Using mixture density networks." *IEEE Robotics & Automation Magazine*, 2020
> >
> > [5] Paraschos, Alexandros, et al. "Probabilistic movement primitives." *Advances in neural information processing systems,* 2013

---

### Official Review · Reviewer_9xkZ · 2022-07-31

**Originality:** Good
**Technical Quality:** Good
**Clarity Of Presentation:** Fair
**Impact:** 2

**Recommendation:**

Weak Reject: I recommend rejecting the paper, but will not argue for my recommendation if the majority of other reviewers have a different opinion.

**Summary:**

This paper proposes an imitation learning approach to capture the multi-modality (versatility) of expert demonstrations. This paper focuses on the trajectory generation aspect of imitation learning while conditioning on relative geometric features such as relative distance to the goal positions.

**Issues:**

- revise the title to be precise about the choice of words.
- missing related works, especially related to the reference that has BC-GMM. And lack comparison to this baseline
- Revise the experiment session to explain the reason of the specific choices of baselines. This also helps to illustrate what the contribution is exactly.
- The last paragraph of the related work is not well-written. The paragraph name needs a more precise summary other than “further relevant approaches”.
- Some grammar issues: “Few modes” -> “a few modes”.
- Explicitly state whether this approach can be applied/extended to pick/place behaviors. If it can, what are the major changes to the approach the authors need? Otherwise, this approach would only be incremental to solve pushing / reaching tasks that are not really complicated not all.

**Quality Of The Limitations Section:**

Limitations are addressed clearly

**Reviewer Expertise:**

5: The reviewer is absolutely certain that the evaluation is correct and very familiar with the relevant literature

**Robotics Focus:**

Sufficient demonstration on hardware

**Strengths And Weaknesses:**

Strengths:
- I can’t tell the strengths of the approach given that it’s missing  a very important baseline, as mentioned in the “weakness” and “issues”

Weakness:
- The work is about learning behaviors / skills / motion primitives, but not a plan technically. The title itself is very misleading.
- It’s missing a very important baseline, behavior cloning + gaussian mixture model (BC-GMM) [1]. This previous approach can outputs GMM actions, and the model is trained with a very simple supervision loss. Solely comparing against vanilla BC is insufficient.
- It’s very vague about the term “context”. What is exactly the context here? Different goals? Then why not just say “goal-conditioned”? There is no clear semantic of “context” in any of the tasks shown. Please be more specific about the definition of “context” (possibly with rigorous definition in the text) or use other terms without ambiguity.
- The choice of comparison baselines is also ambiguous. What is the purpose of comparing against IRL? The authors are not learning rewards for their approach, then why compare against IRL?
- all the tasks involvely only free motion generation (reaching, and pushing but not really involve contact-rich behaviors), and not in the closed-loop manner. How does the approach generalize to pick-place behaviors, and more complicated pushing behaviors?

[1] Mandlekar et al. What Matters in Learning from Offline Human Demonstrations for Robot Manipulation

**Summary Of Recommendation:**

Given that one of the basic baselines is missing to showcase the proposed formulation is indeed necessary to learn versatile behaviors, I vote for weak reject but will change the review if the authors address all my comments and issues properly.

---

> ### Author Response · Authors · 2022-08-22
> **Reply to Reviewer 9xkZ (Part 1/2)**
>
> Please note that this reply has been split into two parts due to character limits. (Part 1/2)
>
> We thank the reviewer for their constructive comments and detailed feedback, as well as for bringing important related work to our attention. In the following, we want to address the individual concerns mentioned by the reviewer. We will upload a revised version of the paper in due time and look forward to further comments from the reviewer.
>
> > The work is about learning behaviors / skills / motion primitives, but not a plan technically. The title itself is very misleading.
> >
>
> We agree that learning a plan often involves some form of cost function, which is not the case in this work. We will adapt the title of the paper to better reflect this distinction.
>
> > It’s missing a very important baseline, behavior cloning + gaussian mixture model (BC-GMM) [1]. This previous approach can outputs GMM actions, and the model is trained with a very simple supervision loss. Solely comparing against vanilla BC is insufficient.
> […]
> >
> >
> > missing related works, especially related to the reference that has BC-GMM. And lack comparison to this baseline
> >
>
> Thank you for pointing us to this work. The method proposed in [1] learns a behavioral cloning policy with a Gaussian Mixture Model as its output using a supervised loss. This baseline is already implemented in our paper, where we named it MBC(S). We also implemented a trajectory-based version called MBC(T), which is taken from [2]. Differences to [1] are that we did not make use of the Low Noise Evaluation Trick, and that during evaluation we generate trajectories independently for each mixture component rather than sampling a new component every step. Table 21 of [1] shows that the Low Noise Evaluation Trick only leads to minor improvements, and thus does not change our results. We also experimented with sampling a mixture component at every step during the evaluation, but did not find a significant difference in performance. We will rename the baselines from MBC(X) to BC-GMM(X) and add a reference to [1] during the revision.
>
> > It’s very vague about the term “context”. What is exactly the context here? Different goals? Then why not just say “goal-conditioned”? There is no clear semantic of “context” in any of the tasks shown. Please be more specific about the definition of “context” (possibly with rigorous definition in the text) or use other terms without ambiguity.
> >
>
> Throughout the paper, we use “context” as a synonym for “task descriptor”. In general, we consider environments for which the goal of an instance of the environment can be described by a (typically low-dimensional) vector. In our experiments, these are the positions of the different targets, and the desired dislocation and rotation of the box. While this concept is closely related to Goal-Conditioned Imitation/Reinforcement Learning, this context can also include something like an obstacle or a starting position. We will add a more concrete definition to the paper.
>
> > The choice of comparison baselines is also ambiguous. What is the purpose of comparing against IRL? The authors are not learning rewards for their approach, then why compare against IRL?
> […]
> >
> > Revise the experiment session to explain the reason of the specific choices of baselines. This also helps to illustrate what the contribution is exactly.
> >
>
> For our baselines, we chose Behavioral Cloning and Generative Adversarial Imitation Learning as those are well-known algorithms that allow for easy comparison of the strengths and weaknesses of our method. We extended the Behavioral Cloning baseline to a Gaussian Mixture Model Policy (named MBC(S) in the paper, as mentioned above), and also considered trajectory-based variants of it for a more thorough comparison.
>
> We chose EM+D-REX as another comparison because it can be seen as a generative alternative to our discriminative approach. Both methods train a model that facilitates expert demonstrations on training contexts to fit novel test contexts, and both methods optimize their policy to match multi-modal distributions over features of these demonstrations. However, EM+D-REX does so by learning a reward function, which is generally less precise and requires careful tuning. Opposed to this, VIGOR iteratively re-trains a discriminator, which allows for a more fine-grained learning signal on test contexts, leading to better results and a generally more robust approach. Experimentally, the robustness to hyperparameters can be seen in the ablation studies in Figures 6/11 for VIGOR, and Figure 12 for EM+D-REX.
>
> We agree that the motivation of the baselines is not well stated in the current version of the paper and will add an explanation of these choices in the revision.

---

> > ### Author Response · Authors · 2022-08-22
> > **Reply to Reviewer 9xkZ (Part 2/2)**
> >
> > Please note that this reply has been split into two parts due to character limits. (Part 2/2)
> >
> > > all the tasks involvely only free motion generation (reaching, and pushing but not really involve contact-rich behaviors), and not in the closed-loop manner. How does the approach generalize to pick-place behaviors, and more complicated pushing behaviors?
> > >
> > > […]
> > > Explicitly state whether this approach can be applied/extended to pick/place behaviors. If it can, what are the major changes to the approach the authors need? Otherwise, this approach would only be incremental to solve pushing / reaching tasks that are not really complicated not all.
> >
> > - The Box Pusher task requires contact-rich manipulation, since the robot needs to apply a significant force to push the box that needs to be compensated for. To show this difference, we will add a figure visualizing the trajectory computed by the policy, and the actual trajectory that is executed on the robot.
> > - VIGOR generates its trajectories by using Movement Primitives, which usually represent a single smooth movement. More complex movements can be achieved by *chaining/sequencing* multiple movement primitives, which has been studied extensively in the Movement Primitive literature [3,4,5]. For example, a pick&place behavior can be achieved by first employing a “(move to and) pick" movement primitive, grabbing the object and then the using a "place" movement primitive to complete the task. These individual movement primitives can be straightforwardly learned within the framework of our approach. We leave this application of VIGOR as a promising direction for future work, and will mention it in the revised paper.
> >
> > > The last paragraph of the related work is not well-written. The paragraph name needs a more precise summary other than “further relevant approaches”.
> > >
> >
> > We agree that the current paragraph name is too non-descriptive and will adapt it to better represent the comparison-based approaches it is about. We will rewrite the paragraph for clarity, and briefly specify the relationship between these approaches and VIGOR.
> >
> > > Some grammar issues: “Few modes” -> “a few modes”.
> > >
> >
> > Thank you for pointing this out! We will fix these issues in the revised version.
> >
> > [1] Mandlekar et al. "What Matters in Learning from Offline Human Demonstrations for Robot Manipulation", *5th Annual Conference on Robot Learning.*, 2021
> >
> > [2] Zhou, You, et al. "Movement primitive learning and generalization: Using mixture density networks." *IEEE Robotics & Automation Magazine*, 2020
> >
> > [3] Daniel, Christian, et al. "Hierarchical relative entropy policy search." *Journal of Machine Learning Research,* 2016
> >
> > [4] Neumann, Gerhard, et al. "Learning complex motions by sequencing simpler motion templates." *Proceedings of the 26th Annual International Conference on Machine Learning,* 2009
> >
> > [5] Manschitz, Simon, et al. "Learning to sequence movement primitives from demonstrations." *IEEE/RSJ International Conference on Intelligent Robots and Systems,* 2014.

---

> > > ### Author Response · Authors · 2022-08-26
> > > **Reply to Reviewer 9xkZ**
> > >
> > > We want to thank the reviewer again for their helpful comments. Since the rebuttal phase is coming to a close, we are grateful for any additional concerns that are not yet addressed by the revised version of the paper.

---

> > > > ### Comment · Reviewer_9xkZ · 2022-08-28
> > > > **Response**
> > > >
> > > > Thanks to the authors for the response and revised pdf. I do see some of my concerns have been addressed. Since the revision of pdf was submitted a little bit late for me to read through carefully, I need to decide the score after careful reading and internal discussion with other reviewers. Several quick comments for the authors to improve the paper nonetheless:
> > > >
> > > > > since the robot needs to apply a significant force to push the box that needs to be compensated for.
> > > >
> > > > It's not clear to readers how heavy the box is. To me the friction is not that bad to perform such a single-stroke pushing behavior.
> > > >
> > > > > For our baselines, we chose Behavioral Cloning and Generative Adversarial Imitation Learning as those are well-known algorithms that allow for easy comparison of the strengths and weaknesses of our method.
> > > >
> > > >  I think authors should choose the baselines to facilitate their arguments that VIGOR helps learn from multi-modal demonstrations, not just being "well-known."

---

### Official Review · Reviewer_24LK · 2022-08-08

**Originality:** Good
**Technical Quality:** Good
**Clarity Of Presentation:** Good
**Impact:** 3

**Recommendation:**

Weak Accept: I recommend accepting the paper, but will not argue for my recommendation if the majority of other reviewers have a different opinion.

**Summary:**

This paper proposes a method for learning skills from multiple diverse demonstrations using an approach that infers multi-modal distributions in a geometric feature space.  It is motivated by skills that can inherently be performed/taught in different ways and a robot may be (and perhaps should be) learning from a dataset that leverages multiple (human) demonstrators.  The method is evaluated against several imitation learning baselines on a reaching task and a pushing task.

**Issues:**

See limitations above.

**Quality Of The Limitations Section:**

Limitations are addressed clearly

**Reviewer Expertise:**

3: The reviewer is fairly confident that the evaluation is correct

**Robotics Focus:**

Sufficient demonstration on hardware

**Strengths And Weaknesses:**

This paper addresses an important, open problem in the Learning from Demonstration (LfD) literature.  It is very relevant in the age of big data, as we can much more feasibly scale up the number of humans to collect data from.  The method proposed is principled and builds from existing literature.  My primary questions are around the evaluation of the method.

Overall strengths and suggestions for improvement are summarised below.

More detailed strengths of paper:
+  The method proposed is mathematically principled and allows for adaptability to multiple learning contexts.  It models a set of demonstrations as multi-modal distributions (Gaussian Mixture Models) in order to capture diversity/versatility in the way humans provide successful demonstrations of the task. Furthermore, it learns multi-modal distributions conditioned upon learning context and trains a discriminator to infer the correct context. This allows the method to account for more diverse contexts.
+ The paper evaluates its method on real robot platforms.
+  Figure 4 (right) is a nice result.  It empirically confirms that the proposed method can better leverage (and hence take advantage of) variation in training contexts, as compared to baseline imitation learning methods tested.

I would suggest the following to improve the paper submission:
-  RE the evaluation: How and why were these particular tasks (reaching and pushing) selected?  What interesting properties does each exhibit with respect to what it highlights about the difficulty of the problem or insights about this method as compared to other imitation learning methods, on inherently multimodal learning tasks?  What hypotheses were being tested with each of the experiments?
-  RE the experimental results shown in Figures 3 & 4: Any insights as to why there was significantly more variance in how the methods performed in the reaching tasks as compared to the pushing task?  For the reaching tasks, I also was a bit unclear. Is the entire demonstration trajectory used for learning or just the final goal position? If the latter, I’m not entirely clear on where the multi-modality in the data is expected to derive from and would request more clarity on this.  In particular, if only the goal position is used as learning signal for the task, what geometric features are expected to vary sufficiently as to induce a multi-modal distribution?
-  Regarding the step of the method that performs distribution matching of behavioural descriptors (subsection 4.2), this seems related to the idea of optimising weights for feature expectation matching in Inverse Reinforcement Learning. I would be interested to understand how these are similar and different?  Or perhaps, does the proposed method build upon feature expectation matching imitation learning approaches?  And if so, in what ways? Why is generalising to geometric representations a difficult challenge?
-  The first sentence under Section 4 (page 4) is causing confusion for me.  Does saying that the approach estimates weights for all training and test configurations imply that the method has access to test configurations at training time?  That would seem like “cheating”, from the perspective of generalising the learned model.  I would recommend clarifying what is intended here.

Additional minor comments:
-  On page 1 (line 30), what does context-insensitive mean in this context?  Is this synonymous with saying context-independent?
-  In the Experiments section (page 5, line 175), why n=5 components? How was this selected?

**Summary Of Recommendation:**

I would recommend accepting the paper, based upon its contribution of a method for addressing inherent multi-modality in skill learning from demonstration.  This is an important and (to my knowledge) currently open problem in the Imitation Learning literature.

---

> ### Author Response · Authors · 2022-08-22
> **Reply to Reviewer 24LK (Part 1/2)**
>
> Please note that this reply has been split into two parts due to character limits. (Part 1/2)
>
> We thank the reviewer for their constructive comments and detailed feedback, and especially for the positive review. In the following, we want to address the individual concerns mentioned by the reviewer. We will upload a revised version of the paper in due time and look forward to further comments from the reviewer.
>
> > RE the evaluation: How and why were these particular tasks (reaching and pushing) selected? What interesting properties does each exhibit with respect to what it highlights about the difficulty of the problem or insights about this method as compared to other imitation learning methods, on inherently multimodal learning tasks? What hypotheses were being tested with each of the experiments?
> >
>
> All tasks were primarily selected due to their multimodal nature and because they allow for an easy collection of human demonstrations.
>
> - The Planar Reacher is used as an introductory task that lends itself well to a qualitative visualization of the underlying multimodality. We show in Figures 3 (left) and 7 that other approaches are unable to find (versatile) solutions to this task.
> - The Panda Reacher task showcases how the different methods deal with high-dimensional problems since the robot is controlled by a total of 48 movement primitive parameters in the trajectory-based experiments.
> - The Box Pusher was chosen to highlight explicit environment interaction. Since we compute the features offline with respect to the initial and desired box position, the approaches need to learn to compensate for the difference between desired and executed trajectory. We find experimentally that only our approach is able to do this in novel contexts, and only requires a few training samples to do so. Additionally, the task shows that VIGOR benefits more heavily from additional data than other approaches.
>
> We agree that this is an important aspect of the task selection and will include a short motivation for each task in the revised paper.
>
> > RE the experimental results shown in Figures 3 & 4: Any insights as to why there was significantly more variance in how the methods performed in the reaching tasks as compared to the pushing task?
> >
>
> We also noticed this during the evaluation of our experiments. Looking at the experiments qualitatively, we find that for the reaching tasks, poor performance often means only pursuing either of the two targets and ignoring the other. This behavior leads to a large distance to at least one target boundary, where the concrete distance depends on which target is pursued and how the robot reaches it. Opposed to this, `failing' at the box pushing task often means plainly missing the box, leaving it at its initial position. This has less impact on the distance metric than a missed target does for the reaching task.
>
> > For the reaching tasks, I also was a bit unclear. Is the entire demonstration trajectory used for learning or just the final goal position? If the latter, I’m not entirely clear on where the multi-modality in the data is expected to derive from and would request more clarity on this. In particular, if only the goal position is used as learning signal for the task, what geometric features are expected to vary sufficiently as to induce a multi-modal distribution?
> >
>
> For all tasks, we use stepwise geometric features of the full trajectory to train the discriminator. An example of this can be seen in the bottom of Figure 1, which shows real expert (left, middle) and learned (right) features over the course of two trajectories each. We will clarify this in the description of the figure.

---

> > ### Author Response · Authors · 2022-08-22
> > **Reply to Reviewer 24LK (Part 2/2)**
> >
> > Please note that this reply has been split into two parts due to character limits. (Part 2/2)
> >
> >
> > > Regarding the step of the method that performs distribution matching of behavioural descriptors (subsection 4.2), this seems related to the idea of optimising weights for feature expectation matching in Inverse Reinforcement Learning. I would be interested to understand how these are similar and different? Or perhaps, does the proposed method build upon feature expectation matching imitation learning approaches? And if so, in what ways? Why is generalising to geometric representations a difficult challenge?
> > >
> >
> > Feature expectation matching approaches such as [1,2] and VIGOR both make use of (potentially hand-crafted) features of the observed states to train their policies. Similarly, they assume that matching the expert demonstration under these features will lead to a well-performing policy. However, feature expectation matching methods generally try to match the *expectations/moments* of the features, whereas VIGOR matches their *distribution* via the reverse KL divergence. This distinction is important for versatile data, as it essentially corresponds to the difference between mode averaging and mode seeking behavior. It also means that VIGOR is not limited to a fixed number of moments, but can match arbitrary distributions. While we would have liked to compare to these feature expectation matching approaches, we are not aware of a method that is easily applicable to non-linear dynamics and multi-modal behavior.
> >
> > Another difference is that these approaches often use linear features, whereas we make use of a neural network as our discriminator to allow for non-linear relationships between the features.
> >
> > If there are any concrete approaches that we should take into account, please let us know. In any case, we will add a brief discussion about these similarities and differences to the paper.
> >
> > > The first sentence under Section 4 (page 4) is causing confusion for me. Does saying that the approach estimates weights for all training and test configurations imply that the method has access to test configurations at training time? That would seem like “cheating”, from the perspective of generalising the learned model. I would recommend clarifying what is intended here.
> > >
> >
> > Thank you for pointing this out! Our approach needs access to the test contexts during training time, but does not see any expert demonstrations from these contexts. The test contexts are needed in advance since the policies learned for these contexts are optimized alongside the policies for the training contexts. However, since we learn from geometric features that can be directly computed from the context and the trajectory proposed by the policy, these trajectories are never executed during training. Hence, for a new test context, we need to rerun the algorithm, which requires computation time but no interaction with the real system.  This procedure is outlined in Figure 2. We currently address this fact in the "re-training" paragraph of the "limitations" section, and will also clarify the beginning of Section 4 and the description of Figure 2 during the revision.
> >
> > > -On page 1 (line 30), what does context-insensitive mean in this context? Is this synonymous with saying context-independent?
> > >
> >
> > The features are not quite context-independent, but constructed in such a way that they generalize well across different contexts. To this end, the features only contain information that abstracts away from the concrete contexts, such as e.g., relative distances instead of absolute positions. We will use the term "concise" instead of "context-insensitive" in the revised paper to make this clearer.
> >
> > > In the Experiments section (page 5, line 175), why n=5 components? How was this selected?
> > >
> >
> > We find that for $n \leq 5$ components, a larger number of components generally leads to improved performance. This can be seen for VIGOR in Figure 6, and for EM+D-REX in the appendix in Figure 12. However, we found experimentally that $n=5$ components are sufficient for matching the distribution of expert features for the considered tasks, and that adding more components generally only leads to spurious improvements.
> >
> > [1] Abbeel, Pieter, and Andrew Y. Ng. "Apprenticeship learning via inverse reinforcement learning." *Proceedings of the twenty-first international conference on Machine learning*, 2004
> >
> > [2] Ziebart, Brian D., et al. "Maximum entropy inverse reinforcement learning." *Aaai*, 2008.

---

### Author Response · Authors · 2022-08-26
**Paper Revision and Changelog (Edit: 27.08.)**

Dear area chair and reviewers,

attached you find a revised version of our paper. For easy reference, we color-coded all changes by the reviewer who suggested them:

- General changes: blue.
- Reviewer **24LK**: purple.
- Reviewer **9xkZ**: green.
- Reviewer **TvrN**: orange.

We briefly list the changes made to the paper, denoting the reviewer(s) and area chair whose comments motivated these changes in parenthesis.

- Unified and simplified notation and improved clarity of presentation throughout the paper.
- Moved the ablation studies to Appendix D for additional space in the main paper.
- Added a “success rate” metric for all approaches and environments in Appendix C (**24LK**).
- Added a brief discussion about the differences and similarities between our approach and feature expectation matching methods (**24LK**).
- Clarified which geometric features are used for which tasks (**24LK**).
- Clarified that our approach needs access to test contexts (but not test demonstrations) during training in the main body of the paper in addition to the “limitations” section (**24LK**).
- Clarified the motivation for the different tasks and added additional descriptions (**24LK**).
- Clarified the motivation behind using $n=5$ mixture components in our experiments (**24LK**).
- Clarified the justification of our choice of tasks and baselines and added additional references where appropriate (**24LK**, **9xkZ**).
- Added a plot showing the difference between planned and executed trajectory induced by the contact with the box for the Box Pusher task (**9xkZ**).
- Mentioned that more complex behaviors can be learned by chaining multiple movement primitives together for a richer movement representation (**9xkZ**).
- Renamed and polished the “further relevant approaches” paragraph of the “related work” section (**9xkZ**).
- Clarified that we use the BC-GMM method introduced in [1] and renamed MBC(X) to BC-GMM(X) accordingly (**9xkZ**).
- Corrected style and grammar issues (**9xkZ**).
- Adapted the title of the paper and general notation to reflect the difference between our approach and the common notion of plans (****9xkZ****).
- Clarified the motivation for our choice of baselines, especially for EM+D-REX (**9xkZ**, **TvrN**).
- Clarified the notation and definition of concepts and mathematical objects (**9xkZ**, **TvrN**).
- Clarified the advantages of the different aspects of our approach (,i.e., trajectory-based mixture policies, distribution matching and an adversarial setup) via a comparison with the baselines (**TvrN**).
- Added a list of contributions and their justification (**TvrN**).
- Clarified the technical discussion (**TvrN**).
- Added a brief discussion on the design and choice of geometric descriptors (**TvrN**).
- Highlighted how the use of geometric descriptors leads to improved sample efficiency (**TvrN**)
- (Edit: 27.08.) Clarified in the "experiments" section that all methods train on the preprocessed expert demonstrations and that the state-action baselines also make use of the geometric descriptors (**TvrN**).

We want to thank the reviewers and the area chair for their helpful comments. We look forward to hearing about any concerns that are left unaddressed by the revised paper, as well as to additional feedback.

[1] Mandlekar et al. “What Matters in Learning from Offline Human Demonstrations for Robot Manipulation”, *5th Annual Conference on Robot Learning*, 2021

---

### Meta-Review · Area_Chair_cokC · 2022-08-15

**Recommendation:** Accept (Poster)
**Confidence:** 3

**Metareview:**

A method is proposed for learning skills from diverse demonstrations, with a focus on inferring
multi-modal distributions and grounded in geometric features.  This allows the method to learn
multiple ways of performing a task, as learned from diverse human demonstrations.

The paper has three reviews, with recommendations of:  weak accept, weak accept, weak reject. Two of the reviewers actively engaged in the discussion.  The reviewers ultimately point to a lack of clarity in positioning the work as being a primary weakness of the paper.
The paper is borderline based on the current reviews and reviewer discussion.

My recommendation is to Accept (Poster), based on importance of the problem (encoding geometric features in a meaningful way into robotic tasks is important, as is multimodality), that a key benchmark had in fact already been included, and that the presentation has now been significantly improved.

Strengths:
- principled approach to an important open problem in learning-from-demonstration
- evaluated on real robot platforms, and with real human demonstrations
- demonstration of advantage of the training variation, as compared to baselines

Weaknesses:
- missing behavior cloning + gaussian mixture model (BC-GMM) baseline comparison, e.g., see
  "What Matters in Learning from Offline Human Demonstrations for Robot Manipulation";
  Comparing against a vanilla BC method is insufficient.  Relatedly, the choice of baselines used
  should be justified  (rebuttal:  this was included, will rename to make it more evident)
- not clear if it generalizes beyond free-motion generation (rebuttal:  demonstrated push tasks are already quite different)
- contribution is somewhat unclear:  needs explicit list of contributions;
  is the use of geometric features part of the claimed novelty?  (rebuttal: provided;  first to use concise geometric features for Adversarial Imitation Learning)
- title may be misleading, as the work is about motion primitives and not really about planning  (new title: "Inferring Versatile Behavior from Demonstrations by Matching Geometric Descriptors")
- paper would benefit from further discussion & insights on various points  (many details added)


**Best Paper Nomination:**

No

---

> ### Author Response · Authors · 2022-08-22
> **Reply to Area Chair cokC**
>
> We thank the area chair for their constructive comments and detailed feedback, and especially for a concise overview of what to address during the rebuttal period. In the following, we want to address the individual concerns mentioned by the area chair. We will upload a revised version of the paper in due time.
>
> > missing behavior cloning + gaussian mixture model (BC-GMM) baseline comparison, e.g., see "What Matters in Learning from Offline Human Demonstrations for Robot Manipulation"; Comparing against a vanilla BC method is insufficient. Relatedly, the choice of baselines used should be justified.
> >
>
> The BC-GMM method proposed in [1] is already implemented as a baseline in our paper, where we named it MBC(S). We also implemented a trajectory-based version called MBC(T). We note that there are two minor differences in the evaluation scheme when compared to [1], and that these differences do not affect our results. We will rename the baselines and add a reference during the revision. For a more thorough discussion, we refer to our comment to the review of reviewer ****9xkZ****.
>
> > not clear if it generalizes beyond free-motion generation
> >
>
> The Box Pusher experiments show that our approach is able to compute accurate desired trajectories for contact-rich pushing behaviors that require a significant amount of force. Here, we observe a significant difference between desired and actual trajectory due to forces required during the contact-rich manipulation of the box. The algorithm implicitly takes these forces into account by learning from desired trajectories (instead of the actual joint trajectories) of the demonstrator. We refer to our answer to reviewer ****9xkZ**** for more details.
>
> > contribution is somewhat unclear: needs explicit list of contributions; is the use of geometric features part of the claimed novelty?
> >
>
> Our list of contributions is as follows:
>
> - We infer desired trajectories from already a small number of multi-modal human expert trajectories by matching a distribution over behavioral descriptors of the desired trajectories to the distribution of the expert.
> - While our approach is similar to other Adversarial Imitation Learning approaches in the sense that it uses a discriminator to update the policy, to the best of our knowledge our approach is the first adversarial approach that utilizes behavioral descriptors instead of state-action distributions to generalize to novel contexts.
> - Other adversarial distribution matching methods can only obtain unimodal behavior and are matching the state-action distribution which does not generalize to new contexts. We detail this in the `Imitation Learning by Distribution Matching' paragraph of the related work section of the paper. Our approach is also different from methods that use mixture policies to match the moments of either state-action pairs [1] or full trajectories [2] rather than using behavioral descriptors.
> - We conduct experiments with human demonstrations as well as on a real robot, performing extensive ablation studies to showcase the importance of different parameter choices and design decisions. The datasets and code will be published with the camera-ready version.
>
> We will add a list of these contributions and their justifications to the revised paper.
>
> > title may be misleading, as the work is about momtion primitives and not really about planning
> >
>
> We agree that learning a plan often involves some form of cost function, which is not the case in this work. We will adapt the title of the paper to better reflect this distinction.
>
> > paper would benefit from further discussion & insights on various points
> >
>
> We will address the concerns regarding the (technical) discussion and further insights in the revised version of the paper.
>
> [1] Mandlekar et al. “What Matters in Learning from Offline Human Demonstrations for Robot Manipulation”, *5th Annual Conference on Robot Learning*, 2021
>
> [2] Zhou, You, et al. "Movement primitive learning and generalization: Using mixture density networks." *IEEE Robotics & Automation Magazine*, 2020